# Clinical and Sociodemographic Profile, Self-Care, Adherence and Motivation for Treatment, and Satisfaction with Social Support in Portuguese Patients with Type 2 Diabetes

**DOI:** 10.3390/jcm13216423

**Published:** 2024-10-26

**Authors:** Virginia M. G. Regufe, Manuel A. Lobão, Natália Cruz-Martins, Carla Luís, Pedro von Hafe, Cristina B. Pinto

**Affiliations:** 1Department of Internal Medicine, Centro Hospitalar e Universitário de São João (CHUSJ), 4200-319 Porto, Portugal; mop15514@mail.telepac.pt; 2Faculty of Medicine, University of Porto, 4200-319 Porto, Portugal; ncmartins@med.up.pt (N.C.-M.); carlaluis@med.up.pt (C.L.); 3School of Economics and Management, University of Porto, 4200-319 Porto, Portugal; lobao.manuel@gmail.com; 4Department of Diagnostic and Therapeutic Technologies, Cooperativa de Ensino Superior Politécnico e Universitário (CESPU), 4585-116 Gandra, Portugal; 5i3S—Instituto de Investigação e Inovação em Saúde, Universidade do Porto, 4200-135 Porto, Portugal; 6Porto School of Nursing, Center for Health Technology and Services Research and Health Research Network (CINTESIS@RISE), 4200-450 Porto, Portugal; cmpinto@esenf.pt

**Keywords:** nursing, type 2 diabetes, self-care, adherence, motivation

## Abstract

**Objective:** This study aimed to characterize the sociodemographic and clinical profiles of Portuguese patients with type 2 diabetes mellitus (T2DM) and to assess their self-care practices, treatment adherence, motivation, and satisfaction with social support. **Methods:** A cross-sectional observational study was conducted at an endocrinology unit in northern Portugal from January 2021 to December 2022. The sample included 303 adult patients with T2DM who provided informed consent. Data were collected using a structured questionnaire addressing sociodemographic, clinical, and self-care aspects. Validated scales were used to assess diabetes knowledge, self-care activities, treatment adherence, motivation, and social support. **Results:** Of the 303 patients enrolled, with a median age of 67 years, 51.2% were female and 68.2% retired. Clinical measures showed a median systolic blood pressure of 135 mmHg, abdominal circumference of 104.6 cm, and BMI of 29.3 kg/m^2^. Self-care practices were suboptimal, with only 25.1% of patients consistently following a healthy diet, and 31% engaged in weekly physical activity. Although treatment adherence was generally high, issues like forgetfulness were reported. Satisfaction with social support varied, with 30% of patients feeling isolated. **Conclusions:** The study identifies significant gaps in diet and physical activity adherence among T2DM patients. There is a need for targeted educational interventions and enhanced support systems to improve self-care and treatment outcomes. Personalized care strategies addressing educational, motivational, and social support factors are crucial to better managing T2DM and improving patient well-being.

## 1. Introduction

Type 2 diabetes mellitus (T2DM), often referred to simply as diabetes, encompasses a diverse group of diseases characterized by hyperglycemia resulting from defects in insulin production, insulin action, or both. It is currently recognized as a global health emergency. According to the International Diabetes Federation (IDF) Diabetes Atlas, in 2021, approximately 537 million adults were diagnosed with diabetes, and if proper measures are not urgently taken to control it, this number could reach 783 million by 2045, representing 12.5% of the world’s population [1]. The significant healthcare burden and the personal and societal impact of diabetes have placed this disease high on the global political agenda, emphasizing the need to reduce epidemic rates and associated risk factors. In addition, the pervasive nature of diabetes underscores its complex interplay with individual lifestyles, societal structures, and healthcare delivery systems worldwide [1].

Diabetes affects people of all ages, genders, ethnicities, and geographic regions, and its development and management are closely linked to people’s lifestyle habits, including diet, physical activity levels, alcohol consumption, smoking, and other behaviors [2]. This group of diseases is also influenced by various social and economic factors, which can exacerbate the risk and management challenges associated with T2DM. For instance, people with low economic power have less access to healthy foods, exercise facilities, and adequate medical care, which increases the risk and difficulty in managing diabetes. Additionally, education and awareness about the disease vary among different social groups [2,3]. On the other hand, healthcare systems around the world vary greatly, whether in terms of resources, accessibility, or quality. Therefore, the efficiency and effectiveness of health systems are crucial to ensure proper early diagnosis, ongoing treatment, and the management of complications associated with diabetes [1,2,3].

In Portugal, the situation is similarly concerning, particularly concerning the prevalence of T2DM among the adult population. According to the Annual Report of the National Diabetes Observatory, published by the Portuguese Diabetes Society, it is estimated that about 1 million Portuguese people are living with diabetes. This number represents approximately 13.6% of the adult population, highlighting the magnitude of the problem in a national context [4]. The complications associated with diabetes contribute significantly to morbidity and mortality, emphasizing the urgent need for effective management and prevention strategies.

The diagnosis of T2DM marks a critical moment, bringing with it significant psychological implications that affect both patients and their families [5]. Diabetes management requires sustained changes in lifestyle and daily routine, essential for self-management after diagnosis. Families play a critical role in providing support and health information and helping people with diabetes cope with the challenges of the condition [6,7]. Because T2DM has a varied clinical profile, often with patients being asymptomatic for long periods, it can result in late diagnoses [7,8]. Thus, recognizing clinical manifestations and chronic complications, such as fatigue, blurred vision, polydipsia, polyuria, unexplained weight loss, frequent infections, peripheral neuropathy, diabetic retinopathy, diabetic nephropathy, cardiovascular disease, and diabetic foot, is crucial for early diagnosis and continuous monitoring [8,9,10].

Owing to the complexity of T2DM, multidisciplinary management including lifestyle modification, adequate glycemic control, regular monitoring, and the treatment of potential complications is essential [11,12]. Regular medical follow-up is crucial to ensure good disease control and to prevent and manage medium- to long-term complications. Recent evidence supports that integrated and personalized strategies can significantly improve health outcomes for patients with T2DM. However, despite the availability of various treatments, such as medication, insulin administration, dietary changes, and regular physical activity, many people with T2DM are unable to maintain adequate blood glucose control, resulting in a low efficacy rate [11,12].

There are many reasons for this ineffectiveness, including the complexity of treatment regimens, a lack of adherence, economic barriers, differences in healthcare systems, and variability in individual response to treatment. In addition, the stress and comorbidities associated with diabetes may further complicate disease management [13]. It is essential to investigate and clarify the reasons for this low effectiveness in order to develop more effective and personalized interventions [11,14]. This includes a better understanding of how social, psychological, and economic factors influence adherence to treatment and overall disease outcomes [11,15]. However, there is a need for comprehensive diabetes treatment programs that not only address medical and lifestyle aspects but also provide psychological support and education tailored to individual needs [15].

In this sense, this study aims to describe the sociodemographic and clinical characteristics of a Portuguese sample of adults diagnosed with T2DM attending an endocrinology clinic, and to examine these findings in relation to self-care, adherence and motivation for treatment, and satisfaction with social support.

## 2. Materials and Methods

### 2.1. Study Design and Sample Selection

A cross-sectional analytical observational study was conducted on a sample of Portuguese patients with type 2 diabetes attending consultations at a reference center in northern Portugal. A non-probabilistic sample was selected, consisting of individuals with T2DM who attended an in-person endocrinology consultation at the hospital between January 2021 and December 2022.

The eligibility criteria for participation in the study were being over 18 years of age, having T2DM, attending the hospital endocrinology consultation, and being able to understand verbally and provide the written informed consent. The exclusion criteria are detailed in the Table 1 according to ICD10 codes.

The sample was sequentially constituted by patients with T2DM who attended a face-to-face endocrinology consultation at the hospital, until the previously defined sample size for a 95% confidence interval and a 10% margin of error was reached, with n = 265, using the methodology described by Agranonik and Hirakata [16].

All subjects gave their informed consent for inclusion before they participated in the study. The study was conducted in accordance with the Declaration of Helsinki of the World Medical Association, considering Law No. 67/98 of 26 October (Helsinki 1964; Tokyo 1975; Venice 1983; Hong Kong 1989; Somerset West 1996; Edinburgh 2000; Washington 2002; Tokyo 2004; Seoul 2008), and the protocol was approved by the Ethics Committee of ULSSJ (nº 310/2020).

### 2.2. Data Collection

To collect data, a questionnaire composed of two parts was used. Part I aimed to gather information on sociodemographic (sex, age, marital status, number of children, geographical area of residence, and educational level), professional (profession and employment status), and clinical data (blood pressure, weight, height, abdominal circumference, HbA1c, body mass index, capillary blood glucose, type of diabetes treatment, associated pathologies, and duration of diabetes). Part II consisted of five questionnaires/scales: Diabetes Knowledge Questionnaire (DKQ-24), validated for the Portuguese population by Bastos [17], the Diabetes Self-Care Activities Scale, validated for the Portuguese population by Bastos, Severo e Lopes [18], the Measure of Treatment Adherence (MAT), validated for the Portuguese population by Delgado and Lima [19], the Treatment Motivation Scale (TMS), validated for the Portuguese population by Apóstolo et al. [20], and the Social Support Satisfaction Scale (ESSS), validated for the Portuguese population by Pais Ribeiro [21].

The questionnaire was always administered by the same researcher to prevent disparities in data collection. The confidentiality and anonymity of the participants were ensured, with each questionnaire being assigned a code composed of four numbers and two letters randomly generated by a computer.

### 2.3. Statistical Analysis

Categorical and continuous variables were collected in this study. Categorical variables were presented as absolute and relative frequencies, and continuous variables as mean and standard deviation, or median and interquartile range (IQR), when applicable. To address the data distribution of continuous variables, the Kolmogorov–Smirnov test was used. Sociodemographic and clinical profile results were correlated with those of self-care activities, adherence and motivation for treatment, and satisfaction with social support, and assessed using the chi-square or Fisher’s exact tests, as appropriate.

All statistical analyses were performed using the Statistical Package for Social Sciences (SPSS, IBM Corp, Chicago, IL, USA) software, version 27.0, with a significance level of 0.05.

## 3. Results

Of a total of 303 patients with diabetes included in this study, with a median age of 67 years, 51.2% were women, whose diagnosis of diabetes had been established for 15 years. Median systolic and diastolic blood pressure were 135 mmHg and 75 mmHg, respectively, and abdominal circumference and BMI were 104.6 ± 11.33 cm and 29.3 ± 4.94 kg/m^2^. Median capillary glucose and glycosylated hemoglobin values were 141.5 and 7.4, respectively, and 78.9% of the patients were currently taking oral antidiabetics and 56.4% insulin. Most patients, respectively, 68.2% and 68.8% are currently retired and married, with half of them (52.8%) having completed the first cycle of literacy, and only 9.3% and 10% having high school and university education (Table 2).

When these patients were assessed with regard to their self-care activities for diabetes, several dimensions were addressed, namely those related to food and specific food intake, physical activity, blood glucose monitoring, and foot care (Table 3). Specifically, when patients were asked about their dietary habits, only 25.1% mentioned they followed a healthy diet in the last 6 days, and 19.2% and 18.2%, respectively, mentioned they did not follow it and followed 5 days/week a dietary plan recommended by a health professional. When specifically asked about the consumption of certain foods, 29.7% reported eating red meat 2 days/week, 21.3% reported eating bread with lunch or dinner, 28.4% reported mixing two or more of the following foods in the same meal (e.g., rice, potatoes, pasta, and beans), and 48.3% and 80.5% reported not consuming alcohol with main meals and outside meals, respectively. In addition, most patients (around 80%) reported eating sweets 0–2 days/week.

Regarding the practice of physical exercise, 31% mentioned not practicing physical exercise in a week and 61.9% not participating in other specific exercise types, such as swimming, walking, and cycling. Around 40% of patients monitored their blood glucose levels on a daily basis, and more than 70% of them reported washing their feet and drying them properly. Concerning adhesion to treatment (Table 4), most patients mentioned they do not forget to take the medication or have been careless with the time of taking it. Similarly, almost all patients never or rarely mentioned having no time to take the medication, taking a drug on their own initiative, or suspending the treatment due to having no medicines at home or another reason.

In terms of treatment motivation (Table 5), more than half of the patients reported always agreeing and almost always agreeing with the reasons that can negatively affect the effectiveness of not taking medication to treat diabetes. Among these, patients reported the presence of personal motivation to take their medication correctly every day, to achieve adequate control of their blood glucose levels within the parameters, and to be actively involved in the management of their disease as key factors. In addition, the fear of failing/disappointing family members and even healthcare professionals accompanying them were also cited as reasons for adherence.

Finally, when patients were asked about their satisfaction with social support (Table 6), their evaluative perspective was more heterogeneous, with about 30% of patients mentioning they feel alone in the world and without support, while others did not; similarly, others mentioned they had reduced their social life as much as they wanted. However, most of them reported being satisfied with their friends, family, and even with the different activities they undertake together, although the feeling of not being able to fulfill all the tasks they want is clearly evident.

On this basis, an attempt was made to identify the main triggers that lead to such responses, in particular the impact of the interplay of both clinical and sociodemographic variables (Figure 1, Figure 2, Figure 3 and Figure 4). With regard to the aspects that motivated adherence to a healthy diet and the consumption of more fruits and vegetables, people taking insulin and also the female sex tended to be more careful than those not taking it (*p* = 0.039), namely with regard to the consumption of bread with main meals (*p* = 0.033), the mixture of rice, potatoes, pasta, and beans in the same meal (*p* = 0.013), and the consumption of alcohol during the meal (*p* < 0.001) and outside it (*p* = 0.005). With regard to blood glucose monitoring, people taking antidiabetic drugs (*p* = 0.048) and insulin (*p* < 0.001) ensured a more strict and daily control of it than those not taking them. Regarding the factors that may affect adherence to treatment, people not taking antidiabetic drugs tended not to have a strict understanding of the real needs to ensure a proper control of blood glucose levels (*p* = 0.048) or even to think they would have no problems stopping taking them on their own initiative (*p* = 0.023).

When analyzing the factors that can influence motivation for treatment, people on insulin treatment are more aware of the importance of their active participation in improving their health (*p* = 0.049), of not disappointing other people (*p* = 0.009), and of the fact that living with diabetes is a challenge for them (*p* = 0.042). In addition, women prefer to follow the guidelines for physical activity rather than thinking about it (*p* = 0.035), and those taking antidiabetic medication believe that following the guidelines is the best thing to do (*p* = 0.048). Finally, when investigating the factors that might influence satisfaction with social support, in general, people using insulin and women were most likely to report feeling alone and without support (*p* = 0.006 and *p* < 0.001, respectively), people using insulin sometimes found it more difficult to find friends to vent to (*p* = 0.008), and women reported not having anyone who understands them and to whom they can vent (*p* = 0.010). People using insulin were also more likely to report wanting to be involved in more organizational activities than they were able to (*p* = 0.022).

Looking specifically at the effect of employment, marital status, and education in this interaction, married people were the most likely to follow a healthy diet (*p* = 0.014), but also to include more rice, pasta, potatoes, and beans in a meal (*p* = 0.026); people with a higher level of education were the most likely to follow a dietary plan recommended by a health professional on more days per week (*p* = 0.028); and retired people were the most likely to consume more fruit and vegetables (*p* = 0.003). On the other hand, people with a higher level of education were the most likely to assess their blood glucose levels (*p* = 0.029), and people with a lower level of education were the most likely to receive recommendations from health professionals to achieve adequate control (*p* = 0.006).

In terms of adherence, retired people were the most compliant, especially in terms of not stopping medication because they felt better (*p* < 0.001), on their own initiative (*p* = 0.049), or because they ran out of medication (*p* = 0.039). On the other hand, in terms of motivation for treatment, retired people were those who reported feeling guilty if they did not follow the doctor’s instructions (*p* = 0.016), wanting the doctor to think they were good patients (*p* = 0. 020), that they would feel ashamed if they did not follow the instructions (*p* = 0.007), that it was better to follow the instructions than to think about them (*p* = 0.010), that such a procedure would be best for them (*p* < 0.001), and that they would feel guilty if they did not follow the instructions (*p* < 0.001). Married people (*p* = 0.020) and those in their first cycle of education (*p* = 0.019) were most likely to report that they exercised because they were afraid others would get bored, and those in their first cycle were also most likely to report that they wanted others to see that they could follow the exercise guidelines (*p* = 0.043).

## 4. Discussion

Clinical data from the patients in this study show a systolic (median of 135 mmHg) and diastolic (median of 75 mmHg) blood pressure within the ranges observed in European studies of hypertension in patients with diabetes [22,23]. Similarly, the waist circumference (mean of 104.6 cm) and BMI (mean of 29.3) overlap with those found in studies that indicate a high prevalence of abdominal obesity among people with diabetes [24,25]. This phenomenon is currently referred to as “diabesity,” characterized by the coexistence of obesity and T2DM, which significantly increases the risk of cardiovascular and metabolic complications. Recent studies have shown that excess visceral adipose tissue promotes insulin resistance and chronic inflammation, contributing not only to the development of T2DM but also to challenges in glycemic control among diagnosed patients [26]. Therefore, diabesity necessitates an integrated clinical approach focusing on both weight reduction and glycemic control, incorporating interventions that include lifestyle modifications, pharmacotherapy, and, in more severe cases, bariatric surgery.

In the sociodemographic context, the sample has a median age of 67 years, with a predominance of women (51.2%) and a high proportion of married patients (68.8%). These characteristics are consistent with other studies that identify diabetes as a prevalent condition in the elderly and predominantly affecting women [27]. However, the low level of education, with only 9.3% and 10% of patients having high school and college education, respectively, is a significant concern, as higher levels of education have been associated with better self-care practices and glycemic control [28]. The study by Jafari et al. [29] emphasizes the importance of interventions that enhance diabetes health literacy among patients, as this can lead to a reduction in stress and burnout, promote self-care skills, and strengthen social support networks, ultimately resulting in an overall improvement in the quality of life for individuals with T2DM.

Socioeconomic inequalities significantly impact diabetes management and self-care practices. For instance, research conducted by Studer et al. [30] revealed that individuals with lower socioeconomic status exhibit higher rates of diabetes-related complications and lower adherence to treatment guidelines. Furthermore, a study by Kahn et al. [31] highlighted that financial barriers, including the cost of medications and medical consultations, are crucial factors preventing many patients from adequately following therapeutic recommendations. Additionally, a lack of health education, often associated with lower socioeconomic status, limits patients’ ability to effectively manage their condition [32]. Thus, a multidimensional approach that includes socioeconomic factors can aid in developing more effective and personalized interventions, thereby improving health outcomes for vulnerable populations.

Adherence to recommended self-care practices revealed significant areas for improvement. Only 25.1% of patients reported consistently following a healthy diet, which is below expectations and recommended practice. European studies show that adherence to diabetes-specific diets can vary but is generally associated with better health outcomes [33]. In a study conducted by Sikalidis and Raboga [34], it was revealed that adherence to self-care activities, including a healthy diet, improved glycemic control and health-related quality of life in patients with T2DM. The lack of adherence to dietary recommendations, such as fruit and vegetable intake and the control of carbohydrate-rich foods, reflects an opportunity for more intensive educational and supportive interventions. Each patient may interpret what constitutes a healthy diet differently, depending on their knowledge, culture, and available resources. Studies indicate that dietary recommendations can be understood in various ways based on factors such as educational level and food culture [35]. Therefore, it is essential for healthcare professionals to tailor nutritional education to the needs and realities of each patient, ensuring that the recommendations are clear and feasible.

Physical activity levels were also below recommendations. With 31% of patients not exercising weekly and 61.9% not participating in specific activities, our results show low adherence to physical activity among people with diabetes, and are consistent with data from other studies [36,37]. Promoting personalized exercise programs and integrating regular physical activity into patients’ daily routines could improve health outcomes. All behaviors related to diet, blood testing, exercise, and foot care influence metabolic control, weight management, quality of life, and the incidence of microvascular complications in patients with T2DM. Therefore, patients should be educated to perform adequate foot care and blood testing to maintain their well-being and prevent disease complications, not just when a health problem has already occurred [38,39].

The results concerning treatment adherence indicate that most patients are diligent about taking their medication. However, some barriers were still reported, such as forgetting to take medication or neglecting schedules. Medication adherence is a critical factor in diabetes management, and strategies to improve adherence often involve personalized interventions and ongoing support [40,41]. Motivation for treatment was found to be robust, with many patients expressing a strong desire to follow medical recommendations and not disappoint family members. This agreement with the literature is crucial, as intrinsic motivation is a key predictor of treatment adherence and health outcomes [42]. Therefore, the need for strategies that increase motivation and address patients’ concerns should be considered to improve treatment adherence.

Satisfaction with social support varies considerably. About 30% of patients felt lonely, reflecting the negative impact of social isolation on diabetes management. Previous studies have shown that social support is essential for treatment adherence and overall well-being in patients with diabetes [43,44]. Evidence suggests that positive social support can improve adherence and health outcomes, highlighting the need for interventions that promote more effective support networks for patients. The analysis of clinical and sociodemographic variables showed that insulin use and the female gender were associated with greater adherence, a healthy diet, and stricter blood glucose monitoring. These findings corroborate the literature that suggests that treatment complexity and gender may influence self-care practices [45].

Educational level also played a significant role, with higher educated patients showing better adherence to dietary recommendations and blood glucose monitoring. The association between education level and self-care practices highlights the importance of targeted educational interventions for lower educated populations [11].

Finally, marital status and variables related to retirement influenced self-care practices and treatment adherence. Married and retired patients exhibited better dietary practices and greater treatment adherence, suggesting that family support and time available for self-care may be important facilitating factors [46,47].

This study contributes significantly to the understanding of self-care practices within a specific population and highlights factors affecting treatment adherence, such as educational level and family support. These findings are consistent with the recent literature that emphasizes the complexity of diabetes management and the necessity for integrated approaches that consider clinical, social, and educational factors to enhance health outcomes in vulnerable populations [26,48]. Nevertheless, it also has limitations. Among these, the relatively restricted geographical sample may limit the generalization of results to other regions or social contexts. Additionally, the study did not account for the impact of the duration of diabetes diagnosis, a factor that can directly influence self-care practices and treatment adherence. Patients diagnosed for longer may demonstrate greater adaptation and experience in managing the disease, while those more recently diagnosed may be in initial adjustment phases [49].

## 5. Conclusions

The results of this study highlight the complexity of diabetes management and the need for multifaceted approaches to improve disease control. Strategies should integrate comprehensive educational programs, robust social support systems, and tailored interventions to improve self-care practices, adherence to treatment, and overall health outcomes. By addressing these factors, healthcare providers can better support patients to manage their condition and mitigate the impact of diabetes on their lives.

## Figures and Tables

**Figure 1 jcm-13-06423-f001:**
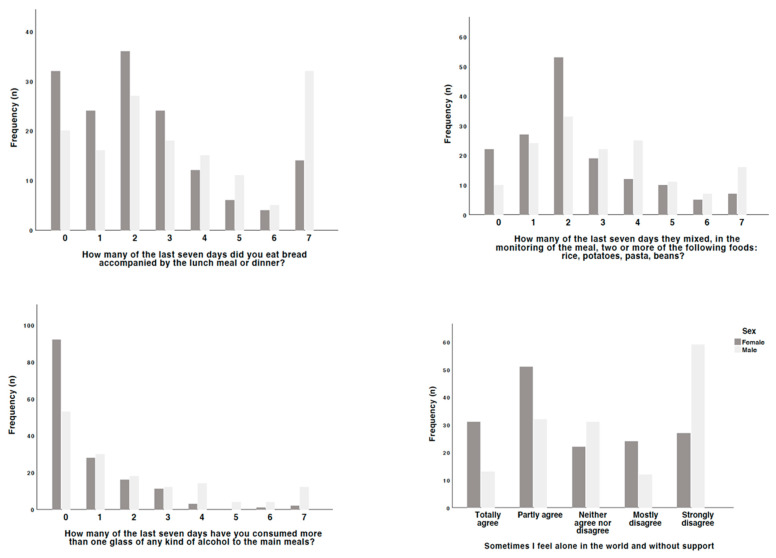
Differences observed by sex in the responses given.

**Figure 2 jcm-13-06423-f002:**
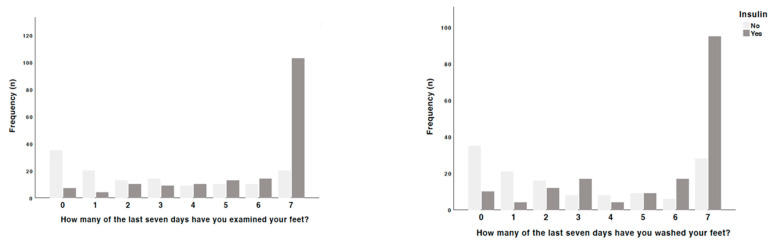
Differences observed among patients taking insulin in the responses given.

**Figure 3 jcm-13-06423-f003:**
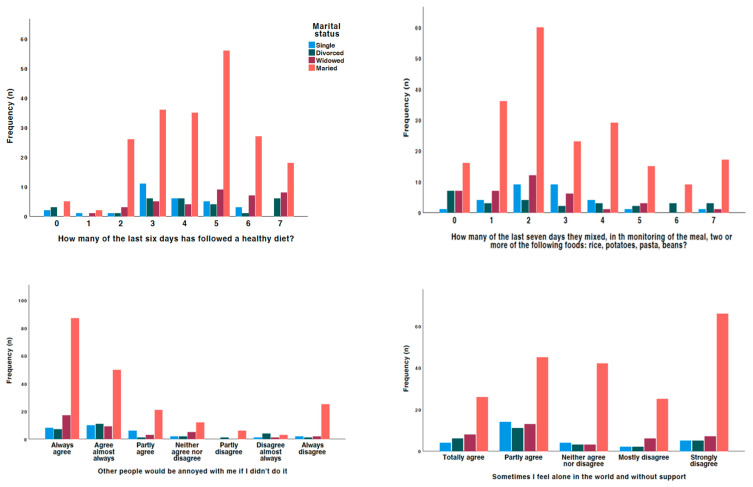
Differences observed by marital status in the responses given.

**Figure 4 jcm-13-06423-f004:**
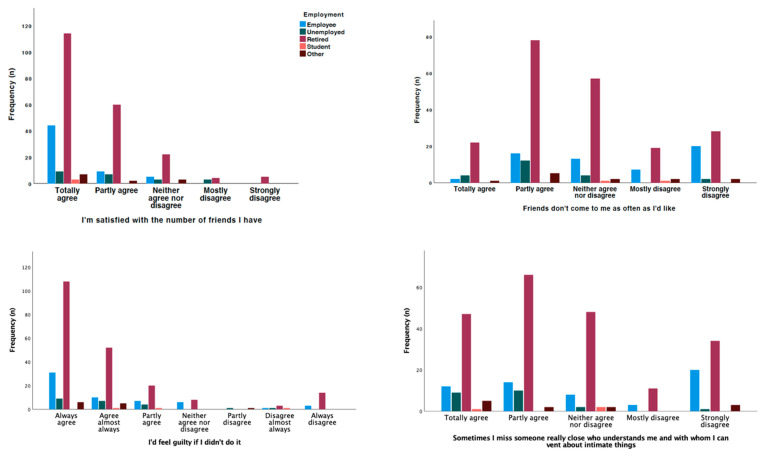
Differences observed by employment status in the responses given.

**Table 1 jcm-13-06423-t001:** Exclusion criteria.

Exclusion Criteria	ICD-10
Oncological Diseases	C00-C97
Degenerative Diseases	G30-G32, I50

**Table 2 jcm-13-06423-t002:** Sociodemographic and clinical features.

Characteristics	N = 303
Age (years), median (IQR)	67 (13)
Systolic arterial tension (mmHg), median (IQR)	135 (23)
Diastolic arterial tension (mmHg), median (IQR)	75 (14)
Abdominal circumference (cm), mean ± SD	104.6 ± 11.33
Glycated hemoglobin, median (IQR)	7.4 (1.7)
BMI (kg/m^2^), mean ± SD	29.3 ± 4.94
Capillary glucose, median (IQR)	141.5 (64.8)
Diagnosis of diabetes (years), median (IQR)	15 (17)
Sex, n (%)	
Women	155 (51.2)
Men	148 (48.8)
Employment, n (%)	
Employee	59 (19.5)
Unemployed	22 (7.3)
Retired	206 (68.2)
Student	3 (1.0)
Other	12 (4.0)
Marital status, n (%)	
Single	29 (9.7)
Divorced	27 (9.1)
Widower	37 (12.4)
Married	205 (68.8)
Schooling, n (%)	
Illiterate	1 (0.3)
1st cycle	159 (52.8)
2nd cycle	49 (16.3)
3rd cycle	34 (11.3)
High school	28 (9.3)
University education	30 (10.0)
Medication used for diabetes, n (%)	
Antidiabetics	239 (78.9)
Insulin	171 (56.4)

**Table 3 jcm-13-06423-t003:** Self-care activities for diabetes.

Number of Days	0	1	2	3	4	5	6	7
Food, n (%)								
How many of the last six days has followed a healthy diet?	10 (3.3)	4 (1.3)	31 (10.2)	59 (19.5)	51 (16.8)	76 (25.1)	39 (12.9)	33 (10.9)
On average, during the last month, how many days a week followed a dietary plan recommended by a health professional?	58 (19.2)	15 (5.0)	26 (8.6)	42 (13.9)	38 (12.6)	55 (18.2)	38 (12.6)	30 (9.9)
How many of the last seven days did you eat five or more fruit and/or vegetable doses (including soup)?	9 (3.0)	12 (4.0)	46 (15.2)	49 (16.2)	36 (11.9)	45 (14.9)	32 (10.6)	73 (24.2)
Specific food, n (%)								
How many of the last seven days did you eat red meat (cow, pig, goat)?	18 (5.9)	29 (9.6)	90 (29.7)	70 (23.1)	40 (13.2)	26 (8.6)	20 (6.6)	10 (3.3)
How many of the last seven days did you eat bread accompanied by the lunch meal or dinner?	52 (17.6)	40 (13.5)	63 (21.3)	42 (14.2)	27 (9.1)	17 (5.7)	9 (3.0)	46 (15.5)
How many of the last seven days they mixed, in the monitoring of the meal, two or more of the following foods: rice, potatoes, pasta, beans?	32 (10.6)	51 (16.8)	86 (28.4)	41 (13.5)	37 (12.2)	21 (6.9)	12 (4.0)	23 (7.6)
How many of the last seven days have you consumed more than one glass of any kind of alcohol to the main meals?	145 (48.3)	58 (19.3)	34 (11.3)	23 (7.7)	17 (5.7)	4 (1.3)	5 (1.7)	14 (4.7)
How many of the last seven days have you consumed any kind of alcohol out of meals?	243 (80.5)	22 (7.3)	13 (4.3)	10 (3.3)	7 (2.3)	0	1 (0.3)	6 (2.0)
How many of the last seven days have eaten sweet foods like cakes, pastilles, jams, honey, marmalade or chocolate?	62 (20.6)	87 (28.9)	93 (30.9)	35 (11.6)	14 (4.7)	4 (1.3)	2 (0.7)	4 (1.3)
Physical activity, n (%)								
How many of the last seven days have practiced physical activity for at least 30 min? (Total minutes of continuous activity, including walking)	94 (31.0)	53 (17.5)	50 (16.5)	35 (11.6)	17 (5.6)	23 (7.6)	12 (4.0)	19 (6.3)
How many of the last seven days participated in a specific exercise session (such as swimming, walking, cycling) beyond the physical activity you do at home or as part of your work?	187 (61.9)	42 (13.9)	17 (5.6)	17 (5.6)	8 (2.6)	11 (3.6)	9 (3.0)	11 (3.6)
Blood glucose monitoring, n (%)								
How many of the last seven days has it evaluated blood sugar?	42 (14.0)	24 (8.0)	23 (7.6)	23 (7.6)	19 (6.3)	23 (7.6)	24 (8.0)	123 (40.9)
How many days a week has you recommended to evaluate blood sugar by your doctor, nurse or pharmacist?	45 (15.1)	25 (8.4)	28 (9.4)	25 (8.4)	12 (4.0)	18 (6.0)	23 (7.7)	123 (41.1)
Feet care, n (%)								
How many of the last seven days have you examined your feet?	23 (7.6)	8 (2.6)	8 (2.6)	17 (5.6)	19 (6.3)	28 (9.2)	51 (16.8)	149 (49.2)
How many of the last seven days have you washed your feet?	1 (0.3)	3 (1.0)	2 (0.7)	7 (2.3)	7 (2.3)	16 (5.3)	46 (15.2)	221 (72.9)
How many of the last seven days did the spaces have dried up between the toes after washing them?	5 (1.7)	3 (1.0)	1 (0.3)	9 (3.0)	8 (2.6)	13 (4.3)	39 (12.9)	225 (74.3)

**Table 4 jcm-13-06423-t004:** Adhesion to treatment.

Questions, n (%)	Always	Almost Always	Frequently	Sometimes	Rarely	Never
Have you ever forgot to take the medication for your illness?	3 (1.0)	2 (0.7)	7 (2.3)	31 (10.2)	117 (38.6)	143 (47.2)
Have you ever been careless with the time of taking your medication?	3 (1.0)	0	11 (3.6)	52 (17.2)	109 (36.0)	128 (42.2)
Has no time to take the medication for your illness because you have better felt?	2 (0.7)	0	2 (0.7)	15 (5.0)	34 (11.3)	249 (82.5)
Has no time to take the medication for your illness, by your initiative, after feeling worse?	1 (0.3)	0	3 (1.0)	10 (3.3)	43 (14.2)	246 (81.2)
Did you take one or more pills for your illness, by your initiative, after feeling worse?	1 (0.3)	0	1 (0.3)	12 (4.0)	33 (10.9)	256 (84.5)
Have you ever interrupted the therapy for your illness because you let your medicines end?	1 (0.3)	0	1 (0.3)	10 (3.3)	55 (18.3)	234 (77.7)
Has no time to take the medication for your illness for any reason other than the doctor’s indication?	2 (0.7)	1 (0.3)	0	9 (3.0)	21 (6.9)	270 (89.1)

**Table 5 jcm-13-06423-t005:** Motivation for the treatment.

**I Take My Diabetes Treatment and/or Control My Glycaemia Because..., n (%)**	**Always Agree**	**Agree Almost Always**	**Partly Agree**	**Neither Agree nor Disagree**	**Partly Disagree**	**Disagree Almost Always**	**Always Disagree**
Other people would be furious with me if I didn’t do it	153 (50.7)	47 (15.6)	19 (6.3)	20 (6.6)	9 (3.0)	10 (3.3)	44 (14.6)
Doing it is a personal challenge for me	138 (45.8)	92 (30.6)	17 (5.6)	16 (5.3)	10 (3.3)	4 (1.3)	24 (8.0)
I believe that doing so will improve my health	167 (55.3)	84 (27.8)	18 (6.0)	8 (2.6)	4 (1.3)	2 (0.7)	18 (6.0)
I would feel guilty if I didn’t do what the doctor said	164 (54.1)	86 (28.4)	18 (5.9)	11 (3.6)	6 (2.0)	4 (1.3)	14 (4.6)
I want the doctor to think I’m a good patient	139 (46.0)	73 (24.2)	37 (12.3)	23 (7.6)	11 (3.6)	2 (0.7)	17 (5.6)
I’d feel bad about myself if I didn’t do it	158 (52.1)	73 (24.1)	38 (12.5)	16 (5.3)	6 (2.0)	1 (0.3)	11 (3.6)
It’s exciting to keep my blood sugar within the recommended range	193 (64.3)	60 (20.0)	23 (7.7)	6 (2.0)	7 (2.3)	1 (0.3)	10 (3.3)
I don’t want other people to be disappointed in me	138 (45.7)	77 (25.5)	38 (12.6)	24 (7.9)	6 (2.0)	3 (1.0)	16 (5.3)
**The Reason I Follow my Diabetes and Exercise Regularly Is Because...**	**Always Agree**	**Agree Almost Always**	**Partly Agree**	**Neither Agree nor Disagree**	**Partly Disagree**	**Disagree Almost Always**	**Always Disagree**
Other people would be annoyed with me if I didn’t do it	120 (39.7)	80 (26.5)	32 (10.6)	21 (7.0)	7 (2.3)	9 (3.0)	33 (10.9)
I believe it’s important to keep me healthier	182 (60.5)	79 (26.2)	12 (4.0)	10 (3.3)	2 (0.7)	3 (1.0)	13 (4.3)
I would feel ashamed of myself if I didn’t do it	145 (48.0)	67 (22.2)	38 (12.6)	28 (9.3)	6 (2.0)	4 (1.3)	14 (4.6)
It’s easier for me to do it than to think about it	161 (53.1)	76 (25.1)	22 (7.3)	20 (6.6)	5 (1.7)	4 (1.3)	15 (5.0)
I’ve given it serious thought and believe it’s the best thing to do	175 (57.8)	71 (23.4)	26 (8.6)	13 (4.3)	2 (0.7)	4 (1.3)	12 (4.0)
I want others to see that I can do it	155 (51.5)	54 (17.9)	22 (7.3)	27 (9.0)	5 (1.7)	13 (4.3)	25 (8.3)
The doctor told me to do it	163 (54.2)	93 (30.9)	19 (6.3)	4 (1.3)	3 (1.0)	7 (2.3)	12 (4.0)
I feel it’s the best thing I can do for myself	164 (54.8)	86 (28.8)	17 (5.7)	6 (2.0)	4 (1.3)	8 (2.7)	14 (4.7)
I’d feel guilty if I didn’t do it	154 (51.2)	76 (25.2)	32 (10.6)	14 (4.7)	2 (0.7)	6 (2.0)	17 (5.6)
They’re the best choices I can make	180 (60.2)	78 (26.1)	15 (5.0)	8 (2.7)	4 (1.3)	3 (1.0)	11 (3.7)
It’s a challenge to learn to live with my diabetes	213 (70.8)	45 (15.0)	12 (4.0)	9 (3.0)	7 (2.3)	4 (1.3)	11 (3.7)

**Table 6 jcm-13-06423-t006:** Satisfaction with social support.

Questions	Totally Agree	Partly Agree	Neither Agree Nor Disagree	Mostly Disagree	Strongly Disagree
Sometimes I feel alone in the world and without support	44 (14.6)	83 (27.5)	53 (17.5)	36 (11.9)	86 (28.5)
I don’t go out with friends as often as I’d like	58 (19.2)	117 (38.7)	61 (20.2)	25 (8.3)	41 (13.6)
Friends don’t come to me as often as I’d like	29 (9.7)	112 (37.5)	77 (25.8)	29 (9.7)	52 (17.4)
When I need to get something off my chest, I can easily find friends to do it with	117 (38.7)	104 (34.4)	45 (14.9)	23 (7.6)	13 (4.3)
Even in the most embarrassing situations, if I need emergency support, I have several people I can turn to	162 (53.8)	86 (28.6)	35 (11.6)	11 (3.7)	7 (2.3)
Sometimes I miss someone really close who understands me and with whom I can vent about intimate things.	74 (24.6)	92 (30.6)	62 (20.6)	15 (5.0)	58 (19.3)
I miss social activities that fulfil me	63 (20.9)	101 (33.6)	84 (27.9)	22 (7.3)	31 (10.3)
I would like to participate more in the activities of organisations (e.g., sports clubs, scouts, political parties, etc.)	62 (20.6)	95 (31.6)	87 (28.9)	12 (4.0)	45 (15.0)
I’m satisfied with the way I relate to my family	174 (57.6)	73 (24.2)	46 (15.2)	4 (1.3)	5 (1.7)
I’m satisfied with the amount of time I spend with my family	158 (52.5)	92 (30.6)	32 (10.6)	14 (4.7)	5 (1.7)
I am satisfied with what I do together with my family	178 (58.9)	83 (27.5)	29 (9.6)	7 (2.3)	5 (1.7)
I’m satisfied with the number of friends I have	178 (59.1)	78 (25.9)	33 (11.0)	7 (2.3)	5 (1.7)
I’m satisfied with the amount of time I spend with my friends	152 (50.3)	97 (32.1)	38 (12.6)	10 (3.3)	5 (1.7)
I’m satisfied with the activities and things I do with my group of friends	157 (52.0)	93 (30.8)	38 (12.6)	10 (3.3)	4 (1.3)
I’m satisfied with the type of friends I have	220 (72.8)	61 (20.2)	15 (5.0)	3 (1.0)	3 (1.0)

## Data Availability

The original contributions presented in the study are included in the article; further inquiries can be directed to the corresponding author.

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
