# Peer review of "Clinical and Sociodemographic Profile, Self-Care, Adherence and Motivation for Treatment, and Satisfaction with Social Support in Portuguese Patients with Type 2 Diabetes"

_jcm, 2024, doi:10.3390/jcm13216423_

Round 1

Reviewer 1 Report

Comments and Suggestions for Authors

Thank you for the opportunity to review the article entitled "Clinical and sociodemographic profile, self-care, adherence and motivation for treatment, and satisfaction with social support in Portuguese type 2 diabetic patients".

The manuscript presents the results of a 2-year cross-sectional observational study aimed at assessing the sociodemographic, clinical and (self-)social profiles of patients with T2DM. This is an important and interesting topic, and although it has been previously exploited in scientific research, I read the article with interest.

The Introduction section briefly introduces the readers to the topic. However, I would like to point out that in lines 36-56 there is no clear information whether the authors are talking about diabetes as a whole or only about T2DM. It is also worth specifying that diabetes is de facto not one disease ("is a type of disease"), but a broad group of diseases characterized by hyperglycemia due to insulin deficiency or improper functioning.

In the methodological section, my possible objections concern the following issue:

1. I suggest consistency in the use of the abbreviation T2D or the type 2 diabetes - despite the previous use of the abbreviation, the full description appears in lines 103, 104, 108 etc.

2. Exclusion criteria are described too briefly. I suggest creating a list or table of disease entities or ICD10 codes that excluded patients from participating in the study.

3. In the section devoted to statistical analysis, the authors do not specify what determines the use of parametric data (mean, SD) and what non-parametric data (median, IQR). Please specify by providing the name of the test used to assess the data distribution.

4. I suggest that the authors add the entire survey referred to in lines 123-135 as a supplementary file.

In the Results section:

1. The authors write about 303 study participants, while in the methodological section it is articulated that the minimum sample size was estimated at 365 patients. Please explain this issue, and if in fact answers were obtained from a smaller number of patients, why was the study time not extended?

2. In Table 1, the authors provide medians and min/max values, while in the statistics section information about providing IQR is also included. Please add the appropriate information, or change your own statistical guidelines.

3. In lines 202-250, the authors draw attention to the search for internal factors that may have an impact on the obtained results. They are described only in text, and only fragmentarily, without providing specific numerical data. I suggest either adding full analyses in tabular form in the supplementary files or (which seems to be an even better option in terms of increasing the attractiveness of the article) adding bar graphs presenting selected differences that are of the greatest importance.

The Discussion section does not raise any major objections. I would suggest expanding it, however - for example, in lines 263-268 you could write a bit more about the phenomenon of diabesity and its clinical significance. There is also a definite lack of information about the limitations and advantages of the study.

Conclusions are beyond doubt.

To sum up, I propose a major revision of the article.

Author Response

Q: Thank you for the opportunity to review the article entitled "Clinical and sociodemographic profile, self-care, adherence and motivation for treatment, and satisfaction with social support in Portuguese type 2 diabetic patients".

A: Thank you for reviewing our manuscript. We appreciate your time and look forward to addressing any suggestions or concerns you may have to improve the quality of the study.

Q: The manuscript presents the results of a 2-year cross-sectional observational study aimed at assessing the sociodemographic, clinical and (self-)social profiles of patients with T2DM. This is an important and interesting topic, and although it has been previously exploited in scientific research, I read the article with interest.

A: We are grateful for your positive feedback on our manuscript. Indeed, the sociodemographic and clinical profiles of patients with T2DM have been extensively studied, but our work aims to provide a detailed look at the Portuguese population, an area with unique characteristics and healthcare challenges. We appreciate your interest in our study and are confident that our findings will contribute valuable insights to the broader body of research on T2DM. Thank you once again for your supportive comments.

Q: The Introduction section briefly introduces the readers to the topic. However, I would like to point out that in lines 36-56 there is no clear information whether the authors are talking about diabetes as a whole or only about T2DM. It is also worth specifying that diabetes is de facto not one disease ("is a type of disease"), but a broad group of diseases characterized by hyperglycemia due to insulin deficiency or improper functioning.

A: We appreciate the valuable comments and suggestions regarding the Introduction section of our manuscript. We fully agree that distinguishing between the different types of diabetes is essential for understanding the topic.

To address your observation regarding the lack of clarity in discussing diabetes in general versus only Type 2 Diabetes Mellitus (T2DM), we have made the following revisions:

In the introduction, we will clarify that the primary focus of our study is Type 2 Diabetes Mellitus, while also acknowledging that the term "diabetes" is often used to describe a broader group of conditions characterized by hyperglycemia. This adjustment will help enhance the accuracy of the information presented.

We have included a clear definition of diabetes as a group of diseases characterized by hyperglycemia, resulting from deficiencies in insulin production or its effective utilization by the body. This addition not only improves clarity but also provides a solid foundation for the subsequent discussion on the impact and management of T2DM.

In the methodological section, my possible objections concern the following issue:

  1. I suggest consistency in the use of the abbreviation T2D or the type 2 diabetes - despite the previous use of the abbreviation, the full description appears in lines 103, 104, 108 etc.

A: We appreciate your insightful suggestion regarding the consistency in the use of the abbreviation T2D or "Type 2 Diabetes." In response to your comment, we have revised the manuscript to ensure that the terminology is consistent throughout the text. We have replaced all terms with "T2DM" to maintain uniformity in nomenclature.

Thank you for highlighting this important aspect, as it improves the clarity and professionalism of our manuscript.

  1. Exclusion criteria are described too briefly. I suggest creating a list or table of disease entities or ICD10 codes that excluded patients from participating in the study.

A: Thank you for your valuable feedback regarding the exclusion criteria outlined in our manuscript. We appreciate your suggestion to enhance clarity and transparency.

In response to your comment, we have revised the methodology section to include a detailed table listing the pathological entities and corresponding ICD-10 codes that led to the exclusion of patients from the study. This addition aims to provide a clearer understanding of the criteria used in our selection process and improve the overall quality of the manuscript.

We believe these modifications address your concerns and enhance the comprehensiveness of our methodology.

  1. In the section devoted to statistical analysis, the authors do not specify what determines the use of parametric data (mean, SD) and what non-parametric data (median, IQR). Please specify by providing the name of the test used to assess the data distribution.

A: Thank you for your valuable comment. Kolmogorov-Smirnov test was used to address data distribution. Such information was added in the statistical analysis section.

  1. I suggest that the authors add the entire survey referred to in lines 123-135 as a supplementary file.

A: Thank you for your valuable suggestion. In response to your comment, we have included the complete questionnaire mentioned in lines 123-135 as a supplementary file. This addition will provide readers with a more comprehensive view of the data collection tool used in our study.

We appreciate your insightful feedback, which has helped to improve the transparency and accessibility of our research.

In the Results section:

  1. The authors write about 303 study participants, while in the methodological section it is articulated that the minimum sample size was estimated at 365 patients. Please explain this issue, and if in fact answers were obtained from a smaller number of patients, why was the study time not extended?

A: We acknowledge that there was an error in the digitization of the sample size number. The minimum estimated sample size was not 365 patients, but rather 265 patients. This mistake has now been corrected in the revised manuscript.

We also confirm that the study included a total of 303 participants, which exceeds the revised minimum sample size. Given that we surpassed the required number of participants, there was no need to extend the study period.

  1. In Table 1, the authors provide medians and min/max values, while in the statistics section information about providing IQR is also included. Please add the appropriate information, or change your own statistical guidelines.

A: Thank you for your valuable comments. As described in the statistical analysis section, median and interquartile range (IQR), minimum and maximum values were used for non-parametric variables where appropriate. We decided to present median and minimum and maximum values instead of IQR (Q1-Q3) to give readers a clearer idea of the range of values obtained. However, if the reviewer prefers to see only the IQR along with the median, we can change the data accordingly.

  1. In lines 202-250, the authors draw attention to the search for internal factors that may have an impact on the obtained results. They are described only in text, and only fragmentarily, without providing specific numerical data. I suggest either adding full analyses in tabular form in the supplementary files or (which seems to be an even better option in terms of increasing the attractiveness of the article) adding bar graphs presenting selected differences that are of the greatest importance.

A: Thank you for your valuable comment. As indicated by the reviewer, we have created 4 figures (Figures 1-4) with the selected differences of the greatest importance.

The Discussion section does not raise any major objections. I would suggest expanding it, however - for example, in lines 263-268 you could write a bit more about the phenomenon of diabesity and its clinical significance. There is also a definite lack of information about the limitations and advantages of the study.

A: Thank you for your valuable suggestions regarding the Discussion section of our manuscript. In accordance with your recommendations, we have expanded the specified section to provide a more comprehensive discussion of the phenomenon of diabesity and its clinical significance. Additionally, we have included a more thorough analysis of the study's limitations and advantages, which we believe are essential for a critical and complete evaluation of the results presented.

The added paragraph is highlighted in yellow for your convenience. We are confident that these revisions enhance the quality of our discussion.

Conclusions are beyond doubt.

A: We appreciate your positive feedback on the Conclusions section. We are pleased that the presentation and clarity of this section were adequate and did not raise any concerns. We will continue to ensure that our conclusions remain concise and well supported by the results obtained.

We remain open to any further suggestions that may improve the manuscript.

Reviewer 2 Report

Comments and Suggestions for Authors

The manuscript submitted by Regufe et al. titled: "Clinical and sociodemographic profile, self-care, adherence and motivation for treatment, and satisfaction with social support in Portuguese type 2 diabetic patients" is an interesting observational human study investigating the self care and treatment adherence for patients with T2DM in Portugal. This is an interesting study which could potentially inform clinical and counseling practice so that there is better quality in the management of T2DM.

The reviewer would like to offer some points below for the authors to consider:

1. It may be helpful to provide some information and statistics on diabetes specifically for Portugal in the introduction.

2. One aspect that is important to consider and perhaps look into in the discussion is the length of diabetes. Patients tend to respond differently depending on the extent of time that they have been diagnosed with the disease.

3. Consider adding the rationale for the sample size selection.

4. BMI does not have units. The kg/m2 is a calculation.

5. As part of the questions asked was on healthy diet. This may be somewhat tricky to interpret. How is healthy diet perceived by each patient. This is a point that would likely need to be discussed separately in the discussion.

6. The discussion is lean, short and with small number of references. The important element here is to compare how the findings in this particular study compare with findings in other settings either within Portugal or in other countries. For example does the locale within Portugal play a role? Would the findings be different if the environment was more urbanized? Did the income play a role? While the authors talk briefly about education they do not address income. 

Here is an example of an article that may be useful towards the discussions of the points raised above: 

  1. Sikalidis AK, raboÄŸa EP (2020) Healthy diet and self-care activities’ adherence improved life-quality and Type 2 Diabetes Mellitus management in Turkish adults. Gazz Med Ital - Arch Sci Med. 179(9):528-37. doi:10.23736/S0393-3660.19.04159-7.

Comments on the Quality of English Language

English is OK but there are some grammatical and syntax errors that would need to be addressed.

Author Response

The manuscript submitted by Regufe et al. titled: "Clinical and sociodemographic profile, self-care, adherence and motivation for treatment, and satisfaction with social support in Portuguese type 2 diabetic patients" is an interesting observational human study investigating the self care and treatment adherence for patients with T2DM in Portugal. This is an interesting study which could potentially inform clinical and counseling practice so that there is better quality in the management of T2DM.

A: Thank you for your positive feedback regarding our manuscript titled "Clinical and sociodemographic profile, self-care, adherence and motivation for treatment, and satisfaction with social support in Portuguese type 2 diabetic patients." We appreciate your recognition of the study's potential to inform clinical and counseling practice in managing Type 2 Diabetes Mellitus (T2DM).

The reviewer would like to offer some points below for the authors to consider:

  1. It may be helpful to provide some information and statistics on diabetes specifically for Portugal in the introduction.

A: We acknowledge your suggestion to include information and statistics on diabetes specifically for Portugal in the introduction. We have added relevant data to provide context for our study and enhance the reader's understanding of the local epidemiology of diabetes. This inclusion aims to underscore the importance of our research within the Portuguese healthcare landscape.

  1. One aspect that is important to consider and perhaps look into in the discussion is the length of diabetes. Patients tend to respond differently depending on the extent of time that they have been diagnosed with the disease.

A: Thank you for your insightful suggestion regarding the consideration of diabetes duration in the Discussion section. We have taken your advice into account and made the necessary adjustments to the text to address how the length of diabetes may impact patient responses.

The revised content is highlighted in yellow for your convenience. We appreciate your input, which has contributed to enhancing the depth of our discussion.

  1. Consider adding the rationale for the sample size selection.

A: The sample size was determined using the methodology described by Agranonik and Hirakata (2013), ensuring a 95% confidence interval and a 10% margin of error. The estimated minimum sample size was 265 participants, which was surpassed with the inclusion of 303 participants in the study. This sample size was deemed sufficient to provide robust statistical power for the analysis of the clinical and sociodemographic variables of interest in this population.

We acknowledge that there was an error in the digitization of the sample size number. The minimum estimated sample size was not 365 patients, but rather 265 patients. This mistake has now been corrected in the revised manuscript.

  1. BMI does not have units. The kg/m2 is a calculation.

A: We appreciate your comment. We have corrected the text to clarify that the Body Mass Index (BMI) is a ratio calculated from weight in kilograms divided by height in meters squared (kg/m²) and does not have its own units.

This adjustment has been made to ensure the accuracy of the information.

  1. As part of the questions asked was on healthy diet. This may be somewhat tricky to interpret. How is healthy diet perceived by each patient. This is a point that would likely need to be discussed separately in the discussion.

A: Thank you for your valuable observation regarding the interpretation of a healthy diet as perceived by each patient. We have considered your suggestion and expanded our discussion to address this important point. The added text highlights the variability in perceptions of a healthy diet among patients and its implications for self-care.

This revised section can be found highlighted in yellow for your convenience. We appreciate your input, which has helped us enhance the clarity and depth of our discussion.

  1. The discussion is lean, short and with small number of references. The important element here is to compare how the findings in this particular study compare with findings in other settings either within Portugal or in other countries. For example does the locale within Portugal play a role? Would the findings be different if the environment was more urbanized? Did the income play a role? While the authors talk briefly about education, they do not address income.

A: We appreciate your feedback concerning the brevity of the discussion and the limited number of references. In response to your suggestions, we have significantly expanded this section to compare our findings with those from other settings, both within Portugal and internationally. We have also included an analysis of how factors such as locale and income may influence the outcomes, alongside our previous discussion of education.

The revised content reflecting these additions is highlighted in yellow for your convenience. Thank you for your constructive insights, which have greatly contributed to the overall improvement of our manuscript.

Here is an example of an article that may be useful towards the discussions of the points raised above:

Sikalidis AK, raboÄŸa EP (2020) Healthy diet and self-care activities’ adherence improved life-quality and Type 2 Diabetes Mellitus management in Turkish adults. Gazz Med Ital - Arch Sci Med. 179(9):528-37. doi:10.23736/S0393-3660.19.04159-7.

A: Thank you for your suggestion regarding the article by Sikalidis and RaboÄŸa (2020). We have consulted the article and incorporated its findings into our discussion. This addition has helped to enrich our analysis of the relationship between healthy diet, self-care activities, and the management of Type 2 Diabetes Mellitus.

We appreciate your valuable input, which has contributed to enhancing the depth of our manuscript.

Round 2

Reviewer 1 Report

Comments and Suggestions for Authors

Thank you for the opportunity to second review the manuscript entitled "Clinical and sociodemographic profile, self-care, adherence and motivation for treatment, and satisfaction with social support in Portuguese type 2 diabetic patients".

The authors have responded to the feedback from the prior review, markedly enhancing the manuscript's quality.

The authors choose to display median and minimum/maximum values rather than the interquartile range (IQR), believing this approach provides a clearer representation of result variability. However, the reviewer contends that the IQR is a superior measure of data variability and recommends reconsideration of this choice.

Apart from the above, I have no additional comments.

Author Response

Thanks for the valuable inputs to our manuscript. Regarding the IQR, as suggested by the reviewer, we have changed the measures accordingly in the manuscript (highlighted in yellow color).

Reviewer 2 Report

Comments and Suggestions for Authors

The authors have addressed reviewer's points reasonably. Proofreading is suggested.

Author Response

Thanks for the positive inputs to our manuscript.